# Role of Oxidative Stress on Insulin Resistance in Diet-Induced Obesity Mice

**DOI:** 10.3390/ijms241512088

**Published:** 2023-07-28

**Authors:** Bruno Luiz da Silva Pieri, Matheus Scarpatto Rodrigues, Hemelin Resende Farias, Gustavo de Bem Silveira, Victória de Souza Gomes da Cunha Ribeiro, Paulo Cesar Lock Silveira, Claudio Teodoro De Souza

**Affiliations:** 1Laboratory of Experimental Pathophysiology, Graduate Program in Health Sciences, University of Southern Santa Catarina (UNESC), Criciúma 88806-000, Brazil; 2Graduate Program in Biological Sciences: Biochemistry, Universidade Federal Do Rio Grande Do Sul, Porto Alegre 90010-150, Brazil; 3Post Graduate Program in Health, Department of Internal Medicine, Medicine School, Federal University of Juiz de Fora, Juiz de Fora 36038-330, Brazil; victoriasgcribeiro@gmail.com

**Keywords:** insulin resistance, obesity, oxidative stress, skeletal muscle

## Abstract

Insulin resistance is the link between obesity and type 2 diabetes mellitus. The molecular mechanism by which obese individuals develop insulin resistance has not yet been fully elucidated; however, inconclusive and contradictory studies have shown that oxidative stress may be involved in the process. Thus, this study aimed to evaluate the effect of reactive species on the mechanism of insulin resistance in diet-induced obese mice. Obese insulin-resistant mice were treated with N-acetylcysteine (NAC; 50 mg/kg per day, for 15 days) by means of oral gavage. Twenty-four hours after the last NAC administration, the animals were euthanized and their tissues were extracted for biochemical and molecular analyses. NAC supplementation induced improved insulin resistance and fasting glycemia, without modifications in food intake, body weight, and adiposity. Obese mice showed increased dichlorofluorescein (DCF) oxidation, reduced catalase (CAT) activity, and reduced glutathione levels (GSH). However, treatment with NAC increased GSH and CAT activity and reduced DCF oxidation. The gastrocnemius muscle of obese mice showed an increase in nuclear factor *kappa* B (NFκB) and protein tyrosine phosphatase (PTP1B) levels, as well as c-Jun N-terminal kinase (JNK) phosphorylation compared to the control group; however, NAC treatment reversed these changes. Considering the molecules involved in insulin signaling, there was a reduction in insulin receptor substrate (IRS) and protein kinase B (Akt) phosphorylation. However, NAC administration increased IRS and Akt phosphorylation and IRS/PI3k (phosphoinositide 3-kinase) association. The results demonstrated that oxidative stress-associated obesity could be a mechanism involved in insulin resistance, at least in this animal model.

## 1. Introduction

The epidemic of obesity is a major public health problem globally. Excess adiposity is a major risk factor in the progression of various metabolic conditions, including type 2 diabetes mellitus (T2DM), dyslipidemia, nonalcoholic fatty liver disease, etc. [1]. These pathological states are strongly associated with insulin resistance. Described as the major link between obesity and T2DM, insulin resistance is a condition in which target peripheral tissues, such as the skeletal muscle, liver, and adipose tissue, have a subnormal response to the levels of circulating insulin, resulting in a decrease in the physiological effects of this hormone and, thus, lower glucose uptake. The precise molecular mechanisms that are involved in the pathogenesis of obesity-associated insulin resistance and its consequences are complex. Based on the efforts over the last decades, there have been remarkable developments in the investigation of obesity-induced insulin resistance, especially in the mechanisms involved in this process [1]. Among these, augmented production and the action of pro-inflammatory cytokines, a process known as subclinical inflammation or low-grade chronic inflammation, is the most widely accepted [1,2,3,4,5,6].

In fact, scientific evidence has revealed that excessive adiposity can increase the levels of pro-inflammatory molecules, such as inductor *kappa* B kinase (IKK), nuclear factor *kappa* B (NFκB), and c-Jun N terminal kinase (JNK), which interpose intracellular insulin signaling, leading to insulin resistance [1,2,3,4,5]. However, it is considered that insulin resistance cannot be explained using one mechanism. Thus, the role of oxidative stress in the mechanism of insulin resistance has been suggested. Radicals derived from oxygen (ROS) and nitrogen (RNS) are the largest class of radical species generated in living systems. ROS and RNS are the products of normal cell metabolism and have either beneficial or deleterious effects, depending on their concentrations in the tissues [7]. A number of studies have reported that ROS and RNS can disrupt insulin signal transduction and have shown elevated levels of ROS in both obese animals and obese humans during insulin resistance [8,9,10,11,12]. Oxidative stress, an imbalance between the production and counteraction of ROS, can increase in certain conditions that generate oxidative stress in obesity, such as hyperglycemia, elevated lipids, vitamin and mineral deficiencies, chronic low-grade inflammation, impaired mitochondrial function, and increased NOX activity [13]. Among these, two principal mechanisms include: elevated production of ROS by the NADPH oxidase (NOX) system [14] and elevated mitochondrial substrate oxidation of fatty acids and glucose [14,15,16,17].

As described above, studies, although not conclusive, have revealed that oxidative stress is the mechanism involved in the onset of insulin resistance. However, there are other studies opposing this [18,19,20]. Ristow et al. observed that exercise-induced oxidative stress ameliorates insulin resistance and causes an adaptive response, promoting endogenous antioxidant defense capacity. However, supplementation with antioxidants precluded these health-promoting effects of exercise in humans [18]. Loh et al. showed that mice lacking one of the key enzymes involved in the elimination of physiological ROS, glutathione peroxidase 1 (Gpx1), were protected from high-fat diet-induced insulin resistance [19]. Considering these controversial and unclear studies on the relationship between insulin resistance and oxidative stress, the present study aimed to evaluate the implications of oxidative stress in insulin resistance in the gastrocnemius of high-fat diet-induced obese mice.

## 2. Results

### 2.1. Biometrics and Physiological Parameters

The results of the physiological evaluations, such as body weight, dietary intake, adiposity index, epididymal fat, fasting glucose, and k_ITT_, are presented in Figure 1. It was observed that the high-fat diet caused a significant increase in the body weight of Swiss mice (HFD group); moreover, treatment with NAC for 15 days did not reduce the body weight of these animals (HFD + NAC) (Figure 1a). Animals in the HFD and HFD + NAC groups showed lower food intake than those in the control group (Figure 1b); however, there was no significant difference when food intake was adjusted by caloric intake (Figure 1c). An analysis of the adiposity index revealed an increase in the percentage of body fat in the HFD group. This did not change after NAC treatment (Figure 1d). On evaluating fasting glucose levels, it was observed that the HFD group had higher fasting blood glucose levels than the control group, and NAC treatment was able to significantly reduce these values (Figure 1e), suggesting improved insulin resistance. In fact, k_ITT_ was reduced in the HFD group, but NAC treatment significantly increased this parameter (Figure 1f).

### 2.2. Oxidants and Antioxidants

Oxidant production was estimated by the oxidation of DCFH-DA. The results suggest that obesity is associated with increased oxidative stress. Figure 2a shows a significant increase in the DCF levels in the HFD group compared to the control group. However, NAC administration significantly reduced the DCF levels. We also evaluated the antioxidant system. Catalase activity was reduced in the HFD group; however, it was increased after NAC supplementation (Figure 2b). In addition, GSH concentration was reduced in the HFD group. However, as expected, NAC treatment was effective in increasing it (Figure 2c).

### 2.3. Inflammatory and Phosphatases Protein Levels

In this study, we evaluated the possible relationship between inflammation and oxidative stress (Figure 3). As expected, the HFD group showed higher NFκB protein levels than the control group; however, NAC supplementation was not effective in reducing NFκB protein levels (Figure 3a). Another inflammatory molecule, JNK, was also evaluated. The HFD group exhibited higher levels of JNK phosphorylation than the control group. However, NAC treatment was able to inhibit this phosphorylation (Figure 3b). Phosphatases may be activated by both inflammation and oxidative stress signals and is involved in the process of insulin resistance. Thus, PTP1B levels were evaluated. A significant increase in PTP1B levels in the HFD group was observed, compared to the control group. NAC treatment was efficient in greatly reducing the PTP1B levels (Figure 3c).

### 2.4. Insulin Signaling Transduction Protein Levels

Next, we evaluated the proteins levels of crucial molecules involved in insulin signal transduction. The HFD group presented lower levels of IRS phosphorylation, compared to the control group. However, treatment with NAC significantly increased the phosphorylation of IRS (Figure 4a). Next, we evaluated the association between IRS and PI3k molecules. The results demonstrated that obesity reduced the immune complex (IRS/PI3k), whereas antioxidant treatment significantly increased IRS/PI3K, compared to the HFD group (Figure 4b). Finally, a significant reduction in Akt phosphorylation was observed in the HFD group, compared to that in the control group. However, the HFD + NAC group showed greater Akt phosphorylation, compared to the HFD group (Figure 4c).

## 3. Discussion

Obesity is a complex metabolic condition, significantly related to T2DM. The results from metabolic and epidemiological studies provide strong evidence that the increasing prevalence of obesity is closely associated with an increase in type 2 diabetes [13]. The crucial link between obesity and T2DM is insulin resistance. Oxidative stress and inflammation are key physiological and pathological events linking obesity, insulin resistance, and the progression of T2DM. However, while there is strong and acceptable evidence of the relationship between insulin resistance and inflammation, the involvement of oxidative stress in this process is contradictory and unclear. Therefore, this study sought to evaluate the effect of oxidative stress on insulin resistance in the gastrocnemius muscle of high-fat diet-induced obese mice. High-fat diet-induced obese Swiss mice were orally supplemented with NAC (a biosynthetic precursor of GSH) for 15 days. The principal results observed were: (a) reduced oxidative parameters; (b) diminished levels of inflammatory molecules, and (c) increased levels of insulin signaling molecules and insulin sensitivity as well as reduced blood glucose levels. These results suggest that oxidative stress might be involved in the process of insulin resistance.

In this study, we treated obese mice with NAC. NAC is an enzymatic antioxidant derived from the amino acid cysteine (chemical formula; C_5_H_9_NO_3_S; molecular weight: 163.2 kDa) [21]. Its antioxidant functions are primarily attributed to its ability to reduce the levels of extracellular amino acid, cysteine, to the intracellular levels. In addition, NAC is an important source of the thiol group. Obese mice treated with NAC showed reduced fasting glucose levels and increased k_ITT_, suggesting the involvement of oxidative stress in the mechanism of insulin sensitivity. Ma et al. (2016) demonstrated the protective effect of NAC on insulin resistance in obese mice. The authors observed that administrating NAC concomitant to a high-fat diet to mice yielded a better result for the ITT and the glucose tolerance test (GTT) compared to the group only fed a high-fat diet. However, these authors observed a reduction in the body weight in these animals; thus, the results may be associated with reduced body weight [22]. In the present study, improvements in the blood glucose and k_ITT_ parameters were observed without changes in body weight and food intake. These results revealed an antioxidant action of NAC on glucose and insulin sensitivity in obese mice, regardless of body weight and adiposity. NAC reducing IR independently of weight loss is clinically interesting, because, for the diabetic patient, losing weight is always difficult.

Next, we used DCF levels as a method to measure oxidative stress [23]. The results showed that obesity elevated DCF levels, suggesting increased ROS. However, NAC treatment was efficient in diminishing DCF levels. Utilizing obese mice (in vivo) and 3T3-L1 adipocytes (in vitro), Houstis et al. observed increased DCF levels after the initiation of insulin resistance [10]. Therefore, after treatment with an antioxidant, namely, MnTBAP, a mimetic of SOD, the levels of DCF were reestablished; moreover, improvements in GTT, ITT, and muscular glucose uptake were observed.

Studies have suggested that elevated oxidative stress in obesity can be attributed to diminished antioxidant systems [7,24]. Therefore, we evaluated CAT activity and GSH concentration. CAT activity was reduced in the HFD group, compared to the control group. NAC treatment increased CAT activity, compared to the HFD group. Abu El-Saad et al. (2016) demonstrated that NAC was effective in elevating CAT activity in hepatic tissue [25]. Barbosa et al. (2013) induced CAT overexpression and observed a diminution in oxidative stress and improved glucose uptake [26]. In the present study, we evaluated GSH levels. GSH, one of the most important and potent antioxidants in the human body, protects against oxidative stress, acting directly by neutralizing various free radical species or by donating electrons to maintain the reduced form of other essential antioxidant enzymes, such as glutaredoxin and glutathione peroxidase [27]. Obese mice showed reductions in GSH levels, compared to the control group. As expected, NAC treatment increased GSH levels, compared to the HFD group.

The ratio of oxidized glutathione to reduced glutathione (GSSG/GSH) is typically used to estimate the redox state of biological systems. A recent longitudinal study, performed by Monzo-Beltran et al., analyzed the GSSG/GSH ratio in morbidly obese individuals subjected to bariatric surgery. The authors observed that obese individuals showed an increased GSSG/GSH ratio, compared to lean subjects [28]. Three months after bariatric surgery, the GSSG/GSH ratio was significantly reduced and remained reduced for one year after surgery. Obese children with insulin resistance demonstrated enhanced levels of oxidative stress biomarkers, such as lipoperoxides and oxidized GSH, in conjunction with stunted antioxidant response after oral glucose loading [29].

A persistent association has been acknowledged among increased adipogenesis, chronic inflammation, and the generation of oxidative stress in the obese state [13]. Oxidative stress stimulates the generation of inflammatory mediators, and inflammation, in turn, enhances the production of reactive oxygen species. Thus, the protein levels of inflammatory molecules, such as NFκB and JNK (Figure 3), were evaluated. NFκB protein levels increased in obese mice, compared to the control group. NAC treatment reduced NFκB protein levels (17.21%); however, significant differences were not observed. However, the protein levels of JNK phosphorylation were greatly reduced after NAC treatment (63.5%), compared to the obese group. As already described previously, the effects of NAC on inflammatory molecules occurs independent of loss of adiposity (Figure 1d). Previous studies observed that elevated levels of ROS activate JNK and NFκB in skeletal muscles [30] and cellular culture [31]. Nouri et al. (2017) showed that NAC treatment in hepatotoxicity-induced oxidative stress in mice was effective in reducing protein levels of inflammatory molecules [32].

After verifying that obesity increases oxidative stress (Figure 2a) and inflammatory molecules (Figure 3a,b), and that NAC treatment improves the parameters of these molecules and increases insulin sensitivity (Figure 1f), we hypothesized that crucial molecules involved in insulin signal transduction could be altered after NAC treatment. In fact, IRS1 and Akt phosphorylation and IRS1/PI3k association increased after NAC treatment (Figure 4). In the present study, NAC was effective in greatly increasing both IRS1 (101.2%) and Akt (122.9%) phosphorylation and IRS1/PI3K association (111.1%). One of the targets of oxidative stress is the activation of multiple serine kinase cascades. These kinases are implicated in the insulin signaling pathway, including the insulin receptor (IR) and insulin receptor substrate (IRS) family of proteins. It has been reported that H_2_O_2_ inhibited insulin-stimulated glucose transport [33,34]. The activation of different signaling pathways, such as NFκB and JNK, appears to be sensitive to oxidative stress and is related to impaired insulin action, suggesting that the paths play a role in oxidative stress-induced IR [33] In adipocytes and muscle cell lines, it was observed that hydrogen peroxide (H_2_O_2_) reduced insulin signaling and, consequently, glucose transport [35]. H_2_O_2_ may induce serine 307 phosphorylation in IRS1, leading to reduced IRS1 protein expression and increased IRS proteolysis [36,37]. In addition to oxidative stress, pro-inflammatory cytokines, including TNFα, can induce serine 307 phosphorylation, thereby. decreasing the interaction of IRS1 and IR, and impairing signal transduction in a similar manner.

Another important mechanism associated with insulin resistance at the molecular level is the level of protein tyrosine phosphatase (PTP1B) [38,39]. In the present study, obese mice presented greatly increased PTP1B protein levels, compared to the control group. In vivo studies demonstrated increased insulin sensitivity and decreased resistance in mice lacking the protein tyrosine phosphatase-1B gene [40]. In humans, single nucleotide polymorphisms of protein tyrosine phosphatase 1B gene were associated with obesity in morbidly obese French subjects [41]. Interestingly, NAC treatment markedly reduced (72.35%) PTP1B levels (Figure 3c), suggesting that increasing protein tyrosine phosphatase could be another (additional to inflammatory molecules) mechanism by which oxidative stress may induce insulin resistance. The mechanism by which oxidative stress modulates PTP1B levels needs further investigation; however, reducing inflammatory status appears to be the way forward. Studies have shown that subclinical inflammation increased the activity of phosphatases, such as PTT1B, and it is the principal mechanism for reducing IR and IRS [42,43,44]. Therefore, in the present study, NAC altered inflammation (Figure 3a,b) and might have had an effect on PTP1B protein levels (Figure 3c), and, accordingly, IRS phosphorylation (Figure 4a) and IRS/PI3k association (Figure 4b).

## 4. Materials and Methods

### 4.1. Animals and Treatment

Eighteen male 4-week-old Swiss mice were randomly divided into two groups. Six mice were fed a standard diet, relative to calories, corresponding to approximately 3.3 kcal/g (Puro Lab 22PB, Porto Alegre, Brazil), as follows: carbohydrate, 70%; protein, 20%; fat, 10%. Twelve mice were fed a high-fat diet, relative to calories, corresponding to approximately 5.35 kcal/g (PragSoluções, Jaú, Brazil) for 12 weeks, as follows: carbohydrate, 26%; protein, 15%; fat, 59%,. After obesity and insulin resistance were established, the animals were redistributed into three experimental groups: control (CNT: standard diet without treatment; *n* = 6), obese group (OB: obese mice without NAC treatment, *n* = 6), and obese group treated with NAC (OB + NAC: obese mice with NAC treatment, *n* = 6). This study was approved by the ethics committee of Extremo Sul Catarinense University, under protocol number 042/2016-2. The administration of NAC (Zanbon, São Paulo, Brazil) was performed daily for 15 days via oral gavage. NAC was administered at a concentration of 50 mg/kg per day in a total volume of 0.5 mL in a single dose [23].

### 4.2. Adiposity Index

After the experimental period, the animals were euthanized (decapitated) and the epididymal, mesenteric, perirenal, and retroperitoneal adipose tissues were extracted and weighed for the calculation of the adiposity index, expressed as the percentage of total body weight (gram of fat/gram of total body weight × 100).

### 4.3. Food and Calorie Intake Measurement

A known amount of food (measured in in grams) was fed to the mice daily. Every 24 hours, the remaining amount of food was weighed. In this way, we were able to roughly estimate the daily food intake of the mice by calculating the difference in the weight of food in two consecutive days. The calorie intake was calculated by converting the grams intake to calories, based on the data offered by the mouse chow provider. So, this amount of measured food intake was multiplied by the nutritional information provided by manufacturers.

### 4.4. Insulin Tolerance Test (ITT)

Fasting insulin tolerance test was conducted 24 h after the last NAC treatment. The first blood collection represented the zero time of the test. Next, Humulin NPH insulin (1 U/kg body weight) was injected intraperitoneally and blood samples were collected through the tail at 5 min, 10 min, 15 min, 20 min, 25 min, and 30 min for the determination of blood glucose. The constant rate of glucose decay (k_ITT_) was calculated using the formula 0.693/t_1/2_. The t_1/2_ of glucose was calculated from the curve of least squares analysis of serum glucose concentration during the linear decay phase.

### 4.5. Dichlorohydrofluorescein Diacetate (DCFH-DA)

Reactive species levels were measured, based on the oxidation of 2′,7′-dichlorodihydrofluorescein (DCF) acetate to the 2′,7′-dichlorodihydrofluorescein fluorescent compound, as previously described [45]. The sample was incubated with 80 mM DCF-DA at 37 °C for 15 min. DCF-DA was de-esterified within the cells by endogenous esterases to free ionized acid, DCFH. DCFH was oxidized to DCF by reactive species. The formation of this oxidized fluorescent derivative was monitored at excitation and emission wavelengths of 488 and 525 nm, respectively. The production of reactive species was quantified using the standard curve of DCF. The results are expressed as units of fluorescence per milligram of protein.

### 4.6. Catalase Activity (CAT)

Catalase (CAT) activity was measured by the rate of decrease in hydrogen peroxide (10 mM) absorbance at 240 nm [46]. To determine catalase activity, tissue samples were sonicated in 50 mM phosphate buffer, and the resulting suspension was centrifuged at 3000× *g* for 10 min. The supernatant was used for the enzyme assay. Enzyme activity is expressed as units per mg of protein. One unit is defined as 1 μmol of reduced hydrogen peroxide per minute.

### 4.7. Glutathione Reduced Levels (GSH)

Glutathione reduced levels (GSH) were determined according to the method described by Hissin and Hilf [47], with modifications. GSH was measured in tissue homogenates after protein precipitation with 10% trichloroacetic acid. An aliquot of each sample was added to 800 mM phosphate buffer (pH 7.4) containing 500 mM DTNB. Color development resulting from the reaction between DTNB and thiols reached a maximum in 5 min and was stable for more than 30 min. Absorbance was determined at 412 nm after 10 min. A standard curve, constructed using reduced glutathione, was used to calculate the GSH levels in the samples. The results are expressed as units of fluorescence per milligram of protein.

### 4.8. Protein Content

The protein content of muscle homogenates was determined using bovine serum albumin as the standard. Folin phenol reagent was added to bind to the protein. The bound reagent slowly reduced, changing from yellow to blue. Absorbance was measured at 700 nm. The results are expressed in milligram.

### 4.9. Western Blot

The gastrocnemius muscles were extracted and immediately homogenized in a specific buffer containing 1% Triton X-100, 100 mM Tris (pH 7.4), 100 mM sodium pyrophosphate, 100 mM sodium fluoride, 10 mM ethylenediaminetetraacetic acid (EDTA), 10 mM sodium vanadate, 2 mM phenylmethylsulfonyl fluoride (PMSF), and 0.1 mg/mL aprotinin at 4 °C using Polytron MR 2100 (Kinematica, Luzern, Switzerland). The homogenate was centrifuged at 11,000 rpm for 30 min at 4 °C. The proteins were resuspended and stored in Laemmli buffer, containing 100 mmol/L dithiothreitol (DTT), until analysis. Further aliquots, containing 125 μg of proteins, were transferred to polyacrylamide gel. Electrophoresis was performed in a Mini-PROTEAN^®^ Tetra electrophoresis system (Bio-Rad, Hercules, CA, USA), with an electrophoresis buffer solution. Proteins separated by SDS-PAGE were transferred to nitrocellulose membranes using Mini Trans-Blot^®^ Electrophoretic Transfer Cell (Bio-Rad) equipment. The nitrocellulose membranes containing the transferred proteins were incubated in blocking solution for 2 h at room temperature to decrease non-specific protein binding. Subsequently, membranes were incubated with the following specific primary antibodies acquired from Cell Signaling Biotechnology (Beverly, MA, USA): NFκB p65, PTP1B, JNK pThr-185/pTyr-183, JNK, IRS1 pTyr-895, PI3k, Akt, Akt pSer-437, and β-actin. In sequence, the membranes were incubated in a solution containing peroxidase-conjugated secondary antibody for 2 h at room temperature. Then, membranes were incubated for 2 min with enzymatic substrate and exposed to RX film in a cassette for development. The intensity and area of the bands were captured using a scanner and, then, quantified through the Scion Image program (Scion Corporation, Frederick, MD, USA). Original membranes were stripped and reblotted with β-actin as the control protein.

### 4.10. Immunoprecipitation

Volumes of samples with 1 mg protein concentrations were used for immunoprecipitation with specific antibodies. The samples were incubated for 12 h at 4 °C under continuous agitation. Then, protein A-Sepharose was added to all samples for precipitation of the antigen/antibody complex and continuously agitated for another 2 h. After centrifugation at 11,000 rpm for another 15 min at 4 °C, the supernatant was discarded and the precipitated material was washed thrice with wash buffer. The precipitated proteins were treated with Laemmli buffer, containing 100 mM DTT, heated in boiling water for 5 min, and centrifuged for 1 min. The samples were then electrophoresed on SDS-PAGE and transferred to nitrocellulose membranes.

### 4.11. Statistical Analysis

The results are expressed as the mean ± standard error of the mean (SEM). The differences between groups were evaluated using one-way analysis of variance (ANOVA), followed by Tukey’s post hoc test. Any *p* values less than 0.05 were considered significant. GraphPad Prism^®^ version 7.0 for Windows was used for data analysis.

## 5. Conclusions

The results of this study demonstrated that oxidative stress-associated obesity could be a mechanism involved in the process of insulin resistance. In addition, NAC seems to improve the IR, independent of weight loss, and dependent on chronic low-grade inflammation reduction.

## Figures and Tables

**Figure 1 ijms-24-12088-f001:**
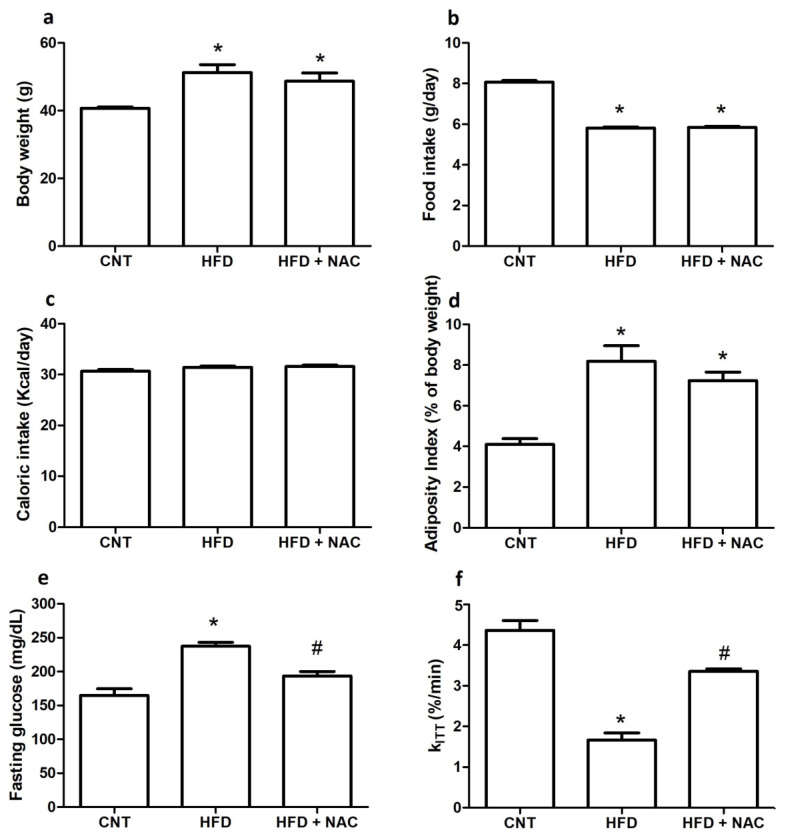
**Effects of NAC administration on the physiological parameters of obese mice.** Body weight (**a**), food intake (**b**), caloric intake (**c**), adiposity index (**d**), fasting glucose (**e**), and constant rate of glucose decay—k_ITT_ (**f**). * *p* < 0.05 compared to the control (CNT) group; # *p* < 0.05 compared to the obese (HFD) group.

**Figure 2 ijms-24-12088-f002:**
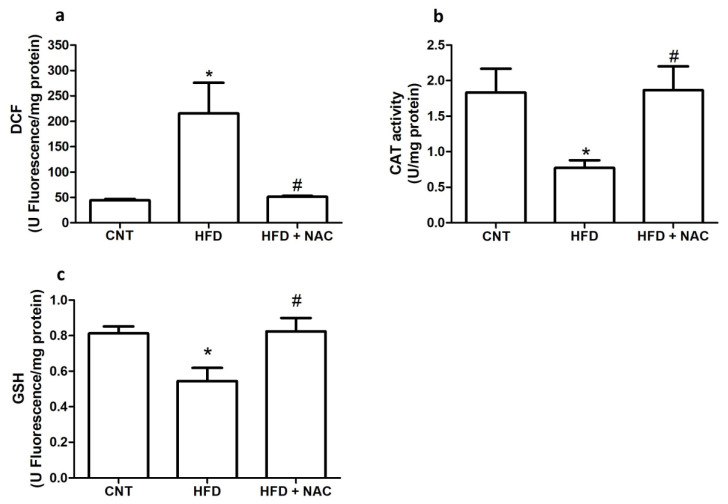
**Effects of NAC administration on the production of oxidants in the gastrocnemius muscle of obese mice.** DCF (**a**), CAT activity (**b**), and glutathione reduced levels (GSH) (**c**). * *p* < 0.05 compared to the control (CNT) group; # *p* < 0.05 compared to the HFD group.

**Figure 3 ijms-24-12088-f003:**
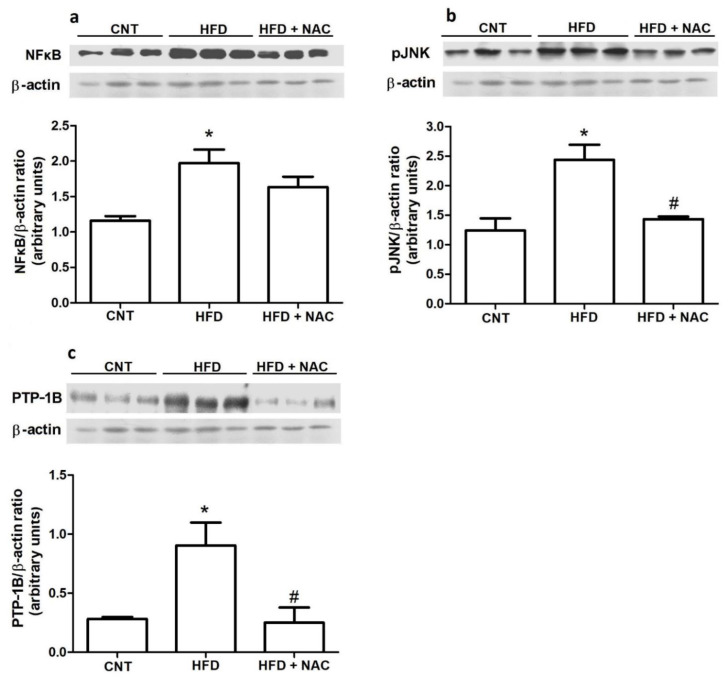
Effects of NAC administration on molecules involved in the inflammatory process in the gastrocnemius muscle of obese mice. Immunoblot analysis of NFκB protein levels (**a**), phosphorylation levels of JNK (**b**), and protein levels of PTP-1B (**c**). * *p* < 0.05 compared to the control (CNT) group; # *p* < 0.05 compared to the HFD group.

**Figure 4 ijms-24-12088-f004:**
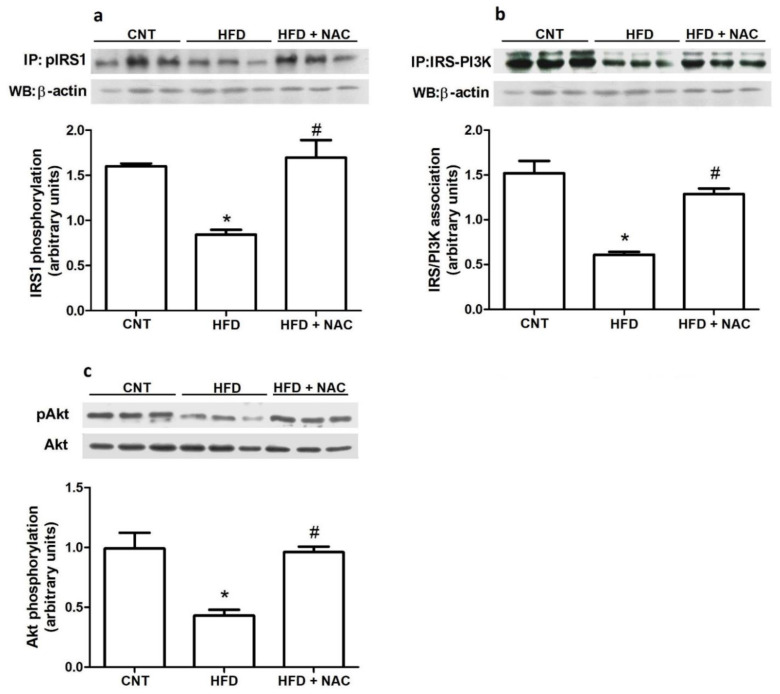
Effects of NAC administration on the expression of molecules involved in insulin signaling in the gastrocnemius muscle of obese mice. Immunoprecipitation for analysis of the association between IRS/phospho tyrosine (**a**) and between IRS/PI3K (**b**), and immunoblot analysis of Akt phosphorylation (**c**). * *p* < 0.05 compared to the control (CNT) group; # *p* < 0.05 compared to the HFD group.

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
