# Peer review of "Role of Oxidative Stress on Insulin Resistance in Diet-Induced Obesity Mice"

_ijms, 2023, doi:10.3390/ijms241512088_

Round 1
Reviewer 1 Report
The study conducted by da Silva Pieri et al. aimed to investigate the relationship between insulin resistance (IR) and oxidative stress in muscle tissue. This was achieved by treating diet-induced obesity (DIO) mice with N-acetyl cysteine (NAC). While previous research has already examined the potential of NAC in alleviating insulin resistance, the authors introduced important findings that contribute to the current understanding in the field. However, the manuscript falls short in effectively accomplishing its main objective as presented. To consider this manuscript for publication, it is crucial for the authors to make specific modifications that address the identified shortcomings and improve the overall clarity and presentation of their work.
The authors failed to provide sufficient commentary in the introduction regarding the use of N-acetyl cysteine (NAC) for reducing insulin resistance (IR). Additionally, there is ample evidence linking IR to the development of obesity, type 2 diabetes mellitus (T2DM), dyslipidemia, and non-alcoholic fatty liver disease (NAFLD) through reactive oxygen species (ROS). To make the manuscript relevant, the authors should distinguish their work from the existing literature by highlighting the unique aspects and contributions of their study.
The authors have not provided information regarding the statistical power of the study and the rationale behind the determination of the sample size. It is essential for the authors to address this aspect and explain their methodology for selecting the number of animals per group to achieve statistical significance. Additionally, to gain a comprehensive understanding of the sample variability, it is crucial for the authors to present all the data, including individual replicates, in the bar graphs. This inclusion will provide readers with a clearer view of the distribution and variability of the results within each group. Furthermore, it would be beneficial if the authors provide the average values and the standard error of the mean (SEM) in all or at least in the most crucial experiments. The authors should conduct statistical analysis comparing each pair of variables in their study (not just against control). Although the authors employed Bonferroni as the post hoc test for multiple variables, it may not be appropriate if all pairs of variables are being analyzed. In such cases, an alternative post hoc test should be used to accurately assess the significance between each pair of variables. I am aware that some data was presented analyzing all pairs of variables but not throughout the study.
Insulin resistance is a multifaceted process that requires thorough examination. However, the authors' presentation of the results falls short in illustrating the continuity of insulin resistance formation. Notably, the authors solely provided body weight measurements at the end of the study, neglecting the progressive changes in body weight over time. To comprehensively assess the impact of N-acetyl cysteine (NAC) on body weight reduction, it is imperative to demonstrate the temporal progression of body weight measurements throughout the study.
In studies assessing insulin resistance (IR) using NAC, it is customary to conduct fasting glucose, oral glucose tolerance tests (OGTT) and insulin tolerance tests (ITT) at both the baseline (before treatment) and the end of the study.
The standard ITT methodology involves measuring glucose levels before insulin injection and at various time intervals such as 15 minutes, 30 minutes, 45 minutes, 60 minutes, and 120 minutes after injection. However, the authors deviated from this established protocol. While the presentation of the insulin sensitivity index (KITT) is acceptable, it is crucial to include the time-dependent curve of the ITT to demonstrate the slope and pattern of the curve over time. This additional information is important for gaining a comprehensive understanding of the ITT results and accurately assessing the impact of NAC on IR.
Western blot showed in figure 3 showed as I understand 3 samples per group. I assumed that the graph is showing at least the 6 replicates of the expression of the protein per group. Again, in is important to show in the bar graph each replicate. The measurements of the b-actin presented in figure 3 and figure 4 are the same. Was b-actin measured again in the IP blots? How many blots per experiment were performed?
Minor: Please provide the site of the phosphorylation of AKT analyzed.
The manuscript is generally well-written, but there are some typos and grammar errors that should be addressed. Additionally, the authors have used various font sizes and styles throughout the document, which should be made consistent. Furthermore, it is noted that most of the citations are placed after the period of each sentence, for example, "...resistance.[1-5]" instead of the correct format "...resistance [1-5]." These issues should be corrected to improve the overall quality and readability of the manuscript.
Author Response
Dear reviewer
We thank you for considering our manuscript for publication in IJMS. We thank you for your advice on our manuscript and would like to resubmit our revised manuscript intitled “Role of oxidative stress on insulin resistance in diet-induced obesity mice” – manuscript ID: ijms-2462778, further consideration.
We have addressed the issues brought up by reviewers. The point-by-point response can be found annexed file. The changes in the manuscript have been highlighted in red.
Best regards

Reviewer 2 Report
1. Materials and Methods:
a) source of diets needs to be indicated.
b) how food and calorie intake were measured?
c) which antibody was used for analysis of NFkB expression?
d)phosphorylated residues need to be indicated for all P-antibodies.
2. Results: a) treatment groups need to be placed above the corresponding lanes on Western blot, Figs 3& 4;
b) authors measured GSH levels, but the ratio of oxidized/reduced glutathione GSH/GSSG is considered as more accurate index of oxidative stress.
3. Discussion: Authors need to state the novelty of this study.
4. Fonts need to be consistent through the manuscript.
Author Response
Dear Editor
We thank you for considering our manuscript for publication in IJMS. We thank you for your advice on our manuscript and would like to resubmit our revised manuscript intitled “Role of oxidative stress on insulin resistance in diet-induced obesity mice” – manuscript ID: ijms-2462778, further consideration.
We have addressed the issues brought up by reviewers. The point-by-point response can be found annexed file. The changes in the manuscript have been highlighted in red.
Best regards

Round 2
Reviewer 2 Report
1.How food and calorie intake were measured- yes, please include it in the text.
2. Lane 304: phospho JNK thr 185/tyr183, JNK, pIRS1 tyr 895, PI3k, Akt, phospho Akt ser437-amino acid residues need to be started with capital letter and it is Akt pSer-473.
Author Response
Dear reviewer,
In response to the comments made by the reviewer, minor revision has been performed. We thank the reviewer for their considerations and suggestions. We believe that the paper has been improved and is now suitable for publication in the IJMS.
Comments and Suggestions for Authors
.How food and calorie intake were measured- yes, please include it in the text.
Response: Yes, it was now included
- Lane 304: phospho JNK thr 185/tyr183, JNK, pIRS1 tyr 895, PI3k, Akt, phospho Akt ser437-amino acid residues need to be started with capital letter and it is Akt pSer-473.
Response: Yes, the reviewer is correct. It was now corrected.
Best regards